# Seroprevalence in Bats and Detection of *Borrelia burgdorferi* in Bat Ectoparasites

**DOI:** 10.3390/microorganisms8030440

**Published:** 2020-03-20

**Authors:** Arinjay Banerjee, Kaushal Baid, Taylor Byron, Alyssa Yip, Caleb Ryan, Prasobh Raveendran Thampy, Hugh Broders, Paul Faure, Karen Mossman

**Affiliations:** 1Michael G. DeGroote Institute for Infectious Disease Research, McMaster Immunology Research Centre, Department of Pathology and Molecular Medicine, McMaster University, Hamilton, ON L8S 4L8, Canada; banera9@mcmaster.ca (A.B.); alyssa.yip14@gmail.com (A.Y.); 2Department of Biochemistry and Biomedical Sciences, McMaster Immunology Research Center, McMaster University, Hamilton, ON L8S 4L8, Canada; baidk@mcmaster.ca; 3Department of Psychology, Neuroscience & Behaviour, McMaster University, Hamilton, ON L8S 4K1, Canada; byront@mcmaster.ca (T.B.); paul4@mcmaster.ca (P.F.); 4Department of Biology, University of Waterloo, Waterloo, ON N2L 3G1, Canada; c9ryan@edu.uwaterloo.ca (C.R.); hugh.broders@uwaterloo.ca (H.B.); 5Department of Veterinary Microbiology, University of Saskatchewan, Saskatoon, SK S7N 5B4, Canada; prr154@mail.usask.ca

**Keywords:** *Borrelia*, big brown bats, seroprevalence, ectoparasite

## Abstract

The role of bats in the enzootic cycle of Lyme disease and relapsing fever-causing bacteria is a matter of speculation. In Canada, *Borrelia burgdorferi* sensu stricto (ss) is the genospecies that is responsible for most cases of Lyme disease in humans. In this study, we determined if big brown bats, *Eptesicus fuscus,* have been exposed to spirochetes from the genus *Borrelia*. We collected serum from 31 bats and tested them for the presence of anti-*Borrelia burgdorferi* antibodies using a commercial enzyme-linked immunosorbent assay (ELISA). We detected cross-reactive antibodies to *Borrelia* spp. in 14 of 31 bats. We confirmed the ELISA data using a commercial immunoblot assay. Pooled sera from ELISA-positive bats also cross-reacted with *Borrelia* antigens coated on the immunoblot strips, whereas pooled sera from ELISA-negative bats did not bind to *Borrelia* spp. antigens. Furthermore, to identify if bat ectoparasites, such as mites, can carry *Borrelia* spp., we analyzed DNA from 142 bat ectoparasites that were collected between 2003 and 2019. We detected DNA for the *Borrelia burgdorferi flaB* gene in one bat mite, *Spinturnix americanus*. The low detection rate of *Borrelia burgdorferi* DNA in bat ectoparasites suggests that bats are not reservoirs of this bacterium. Data from this study also raises intriguing questions about *Borrelia* infections in bats, including the role of humoral immunity and the ability of bats to be infected with *Borrelia burgdorferi*. This study can lead to more sampling efforts and controlled laboratory studies to identify if bats can be infected with *Borrelia burgdorferi* and the role of bat ectoparasites, such as *S. americanus,* in the transmission of this spirochete. Furthermore, we outlined reagents that can be used to adapt ELISA kits and immunoblot strips for use with bat sera.

## 1. Introduction

Numerous emerging pathogenic viruses have been detected in bats [1], but few studies have looked at the prevalence of bacteria that can cause disease in other mammals, including humans. Lyme disease is a tick-borne disease caused by spirochete bacteria belonging to the *Borrelia burgdorferi* sensu lato (sl) genospecies complex [2,3]. In North America, these tick-borne bacteria are transmitted to vertebrate hosts, including humans, through the bite of Ixodes ticks. In Canada, *Borrelia burgdorferi* sensu stricto (ss) is the genospecies that is responsible for most cases of Lyme disease in humans [4]. Other species, such as *Borrelia bissettii, Borrelia kurtenbachii*, and *Borrelia mayonii*, are less commonly associated with human disease.

Rodents and birds are the reservoir hosts that support the enzootic cycle of *B. burgdorferi* sl in nature [5,6]. In addition to the *B. burgdorferi* sl group, there are other species of *Borrelia* that can cause relapsing fever [7]. Although the role of bats as reservoirs of relapsing fever-causing bacteria was speculated upon [8] and sequences of *Borrelia* spp. were identified in bat ectoparasites [9,10,11,12], a limited number of studies looked at the seroprevalence of *B. burgdorferi* in insectivorous bats, along with detecting *B. burgdorferi* DNA in their ectoparasites.

In this study, we sought to identify if bats in Canada were exposed to *B. burgdorferi* spirochetes. Furthermore, archived samples of bat ectoparasites were used as surrogates to determine if bats could potentially be exposed to *B. burgdorferi* via vectors.

## 2. Methods

### 2.1. Sample Collection (Bats and Ectoparasites)

Between 2012 and 2018, 31 big brown bats (*Eptesicus fuscus*) were collected and maintained in a captive colony (see Appendix A for details on location, year bats were captured, and seropositivity status). All visible ectoparasites were removed and bats were treated with topical selamectin (10–20 μL, Revolution^©^ purple for dogs; 120 mg/mL) prior to introduction to the colony. Blood was collected from the interfemoral vein of healthy big brown bats (*n* = 31) and serum was separated by centrifuging the blood at 1500 g (Beckman Coulter, Brea, CA, USA) for 15 min. Serum was aliquoted and frozen at −80 °C prior to analysis. Blood was also collected from bats that were born in captivity within this colony (Bat IDs: 13 pink, 189 white, 31 green, 8 pink, 50 green, 10 pink, 54 blue, 39 gray, 57 sky, 26 green, and 108 red) to monitor exposure within the colony. Bats born in captivity were categorized under Hamilton for location. Ectoparasites (*n* = 142) were collected from *Myotis lucifugus*, *E. fuscus,* and *Myotis septentrionalis* from Eastern Canada between 2003 and 2019 (see Appendix A for demographic details of the samples, reproductive status and gender of bats, and the species of ectoparasite collected from bats). Animal protocols for bat handling were approved by McMaster University’s and Saint Mary’s University’s animal research ethics boards. For sample details, see Appendix A.

### 2.2. Enzyme-Linked Immunosorbent Assay (ELISA)

To detect antibodies against *B. burgdorferi* in *E. fuscus* serum, a commercial ELISA (GenWay Biotech, San Diego, CA, USA) was used. As described by the manufacturer, the ELISA plates were coated with *Borrelia burgdorferi* antigens. Samples were diluted 1:101 and the assay was performed following the manufacturer’s recommendations. All samples were assayed in duplicate. To detect bat immunoglobulin G (IgG) that bound to *B. burgdorferi* antigens, 5 μg/mL polyclonal goat anti-bat IgG labelled with horseradish peroxidase was used (anti-bat IgG-HRP; Bethyl Laboratories Inc., Montgomery, TX, USA). Control human samples were detected using an anti-human HRP conjugate that was supplied with the kit (GenWay Biotech, San Diego, CA, USA). An anti-bat IgG-HRP-only control and a substrate control were included with every assay to account for non-specific absorbance. Absorbance values for bat serum samples were normalized to both the anti-bat IgG-HRP control and the substrate control. Human control samples were normalized to the substrate control as recommended by the manufacturer.

### 2.3. Immunoblots

To detect anti-*Borrelia* antibodies in bat (*E. fuscus*) sera, immunoblots were performed using commercially available strips that were coated with antigens from *B. burgdorferi* sensu stricto (Bb), *Borrelia afzelii* (Bf), and *Borrelia garinii* (Bg) (EUROIMMUN, Lubeck, Germany). For antigen description, see the manufacturer’s website. The manufacturer’s recommended procedure was followed for the assay, but the secondary antibodies and the detection chemistry were altered to adapt the kit for bat sera. Briefly, the strips were incubated with 1.5 mL of 0.1× human positive control (supplied as a 50× concentrate). A quantity of 1.5 mL of diluent was used (universal buffer; EUROIMMUN, Lubeck, Germany) as negative human control. For bats, sera were pooled from ELISA positive or negative bat samples (see Appendix A) in 1.5 mL of universal buffer. After blocking the strips in 1.5 mL of universal buffer for 15 min, 1.5 mL of the respective sera were added on the strips in 15 mL conical screw cap tubes. The tubes were gently rocked on a rocking platform for 1.5 h at room temperature. After 1.5 h, sera were removed and the strips were washed three times in 1.5 mL of universal buffer for 5 min each on a rocking platform. After the washes, strips that had been incubated with human sera were incubated in a 1:10,000 dilution of goat anti-human IgG-HRP conjugate (Millipore, Burlington, MA, USA; catalogue number: AP112P) and strips that had been incubated with bat sera were incubated with a 1:5000 dilution of goat anti-bat IgG-HRP conjugate (Bethyl laboratories Inc., Montgomery, TX, USA; catalogue number: A140-118P). All antibodies were diluted in 1.5 mL of universal buffer. The strips were incubated in the diluted antibodies for 1 h at room temperature on a rocking platform. The strips were then washed three times, as mentioned above, and processed for signal development using a non-commercial enhanced chemiluminescence (ECL) solution [13].

### 2.4. Bacterial Isolation and Detection

A BSK-H media with 6% rabbit serum (Sigma-Aldrich, St. Louis, MO, USA) was used for bacterial isolation. Blood from *E. fuscus* bats were pooled and inoculated in two 15 mL tubes, each containing 6 mL of BSK-H media. The tubes were incubated at 33 °C with 5% CO_2_ in a humidified incubator for 8 weeks. Fresh media was added to the culture every week and the cultures were sub-cultured every two weeks. Aliquots were processed for gram staining (Fischer Scientific, Waltham, MA, USA) and immunofluorescent staining using rabbit anti-*Borrelia burgdorferi* antibody (BioRad, Hercules, CA, USA) every two weeks. Gram staining was carried out following the manufacturer’s recommendations (Millipore Sigma, Burlington, MA, USA). Immunofluorescent staining was carried out as previously mentioned [14]. For immunofluorescent staining, a 1:100 dilution of the primary antibody (rabbit anti-*Borrelia burgdorferi*) and a 1:400 dilution of the secondary antibody (Donkey anti-rabbit Alexa Fluor 488; Life Technologies, Carlsbad, CA, USA) were used. Slides were observed using a Zeiss fluorescent microscope. *Cpn60* sequencing of the total bacterial population was performed, as previously mentioned [15].

### 2.5. Polymerase Chain Reaction (PCR)

DNA extractions were performed from ectoparasites, as previously mentioned [16]. The ectoparasites were classified by amplifying a 658 bp fragment of the cytochrome oxidase subunit I (COI) gene using polymerase chain reaction, as previously mentioned [17]. Briefly, PCR was performed with Q5 high-fidelity DNA polymerase, using the manufacturer’s recommended protocol (New England Biolabs Inc., Ipswich, MA, USA). 5 uL of DNA was used as template. Primers LCO1490: 5’-GGTCAACAAATCATAAAGATATTGG-3’ and HCO2198: 5’-TAAACTTCAGGGTGACCAAAAAATCA-3’ were used to amplify the COI gene. For amplification, an initial step at 98 °C for 30 s was followed by 40 cycles of denaturation at 98 °C for 10 s, annealing at 56 °C for 10 s, and elongation at 72 °C for 1 min. Final extension was performed at 72 °C for 2 min. DNA from *Borrelia burgdorferi* was detected by amplifying a 370 bp fragment of the *flaB* gene using a previously published nested PCR protocol [18]. Briefly, *Borrelia* genus specific primers [18] were used for nested PCR. Primers used for nested PCR-1 were FO1: AAGTAGAAAAAGTCTTAGTAAGAATGAAGGA and FO2: AATTGCATACTCAGTACTAT TCTTTATAGAT. Primers used for nested PCR-2 were FI1-CACATATTCAGATGCAGA CAGAGGTTCTA and FI2: GAAGGTGCTGTAGCAGGTGCTGGCT GT. For amplification, an initial step at 94 °C for 3 min was followed by 35 cycles of denaturation at 94 °C for 30 s, annealing at 50 °C for 30 s, and elongation at 72 °C for 1 min. Final extension was performed at 72 °C for 10 min.

### 2.6. Data Analysis

The commercial ELISA kit contained control samples to determine human cut-off values. However, to determine the cut-off values for bat serum samples, these controls could not be utilized. We determined the mean absorbance value of negative serum samples (Bat IDs: 8 pink, 50 green, 54 blue, 39 gray, 58 sky, 97 purple, 26 green, 19 sky, 38 blue, and 88 yellow) and established a cut-off value of 3 standard deviations (i.e., a cut-off absorbance value of 0.475) to identify bat serum samples that were positive.

### 2.7. Phylogenetic Tree

Multiple sequence alignment and phylogenetic trees were constructed using MEGA 7 (version 7.0.26, https://www.megasoftware.net, accessed on 10^th^ September, 2019) [19].

## 3. Results

*Eptesicus fuscus* bats were collected from different areas in Ontario between 2012 and 2018 and maintained in a research colony (Figure 1A). To detect IgG in bat sera that bound to *B. burgdorferi* antigens, we analyzed bat sera and supplied human control sera using a commercial ELISA kit. We detected *B. burgdorferi* cross-reactive antibodies in 14 of 31 (45.16%) bats (Figure 1B). Sera from 4 of 11 (36.36%) bats born in captivity also tested positive (Appendix A).

Next, immunoblot analyses were performed using commercially available strips that were coated with antigens from *B. burgdorferi* ss, *B. garinii*, and *B. afzelii*. Pooled sera from ELISA-positive bats cross-reacted with multiple antigens, including p41 (flagellin), OspC, and p18, whereas pooled sera from ELISA-negative bats did not cross-react with antigens on the immunoblot strip (Figure 1C). Although we attempted to culture *Borrelia* spp. from bat blood, we were unable to detect spirochetes in the culture by immunofluorescence microscopy or *cpn60* sequencing of the bacterial population.

To detect the *B. burgdorferi flaB* gene in bat ectoparasites (Figure 1D and Appendix A), DNA extracted from ectoparasites collected from *Myotis lucifugus*, *E. fuscus*, and *Myotis septentrionalis* from Eastern Canada between 2003 and 2019 were analyzed. A 370 bp fragment of the *B. burgdorferi flaB* gene was amplified from one bat mite, *Spinturnix americanus,* that was collected from *Myotis lucifugus* in 2011 (the sequence information has been submitted to the GenBank database under accession number MN954474) (Figure 1E and Appendix A).

## 4. Discussion

Bats are speculated to be reservoirs for several viruses [1], including tick-borne viruses [20]. Recently, the role of cave-dwelling bats and Ixodes ticks in the life cycle of *B. burgdorferi* sl was demonstrated in Romania and Poland [10]. Seroprevalence data from our study demonstrate that wild-caught and captive-bred Canadian bats have been exposed to *B. burgdorferi* or an antigenically related bacterium. Our assays were not designed to specifically detect antibodies to relapsing fever-causing spirochetes in bats; however, the presence of cross-reactive antibodies against antigenically similar spirochetes cannot be ruled out. Since wild-caught bats and bats born in captivity were either positive or negative within the same colony, it is unlikely that ectoparasites transmitted the bacterium within the research colony, but this possibility cannot be ruled out. These data raise intriguing questions about the possibility of vertical transmission of *Borrelia* spp. in bats, individual- or population-level persistence of infection, multiple exposures to *Borrelia* spirochetes, and the role of maternal antibodies in juvenile bats. Humoral immunity in bats is poorly understood [21]. As bat reagents and tools become available, it will be interesting to explore aspects of bat humoral immune responses against parasitizing bacteria.

Despite multiple attempts, we were unable to detect *Borrelia* DNA in bat blood cells using a nested PCR approach for multiple targets (16s rRNA, 5s-23s rRNA intergenic spacer, *flaB* gene; data not shown). In addition, we were unable to culture spirochetes from *E. fuscus* blood. This is not surprising, since *Borrelia* spirochetes are likely to be present in very low numbers in an asymptomatic host. Furthermore, blood may not be a suitable sample; more invasive sampling, including tissues, may be required to isolate any *Borrelia*-like spirochetes from bats.

*B. burgdorferi* sl was previously detected in mites from the genus *Laelaps* [22]. We detected a *B. burgdorferi flaB* gene segment in a bat mite, *Spinturnix americanus*. The role of *S. americanus* in acquiring and transmitting *Borrelia* spirochetes in bats needs to be investigated. From our data, the possibility that bats in Canada can acquire the spirochete cannot be ruled out. Furthermore, detecting *B. burgdorferi* DNA in a bat mite does not imply transmission or acquisition of the spirochete to and from bats, respectively. As discussed below, controlled laboratory experiments are required to establish the acquisition and transmission potential of *B. burgdorferi* by bat mites. Larger numbers of ectoparasites from bats, including *E. fuscus* in Ontario, need to be investigated to support our sero-surveillance data. Collecting additional samples from the sampling site of ectoparasite 8875 might allow for the identification of any localized spread of *Borrelia* spp. It is also possible that bats are exposed to *Borrelia* spp. through a different route, but do not sustain an infection for prolonged periods of time, thus limiting the opportunity for bat ectoparasites to acquire the spirochete through a blood meal. Surveillance data from bat ectoparasites can enable researchers to rationalize the development of *in vivo* experiments with bats and their ectoparasites in order to fully dissect the ability of bat mites to acquire and transmit *B. burgdorferi* within bat populations and to other mammalian species, such as rodents.

In our attempt to identify cross-reactive *Borrelia burgdorferi* antibodies in bats, we optimized and developed methods to adapt existing commercial ELISA kits and immunoblot strips for use with bat sera. These techniques and information about anti-bat secondary antibodies can allow other researchers to adapt and utilize commercial kits for the study of bat sera.

Bats play an important ecological role and carry out vital functions, such as pollination, insect population control, and seed dispersal. It is important to study and reduce human–bat interactions to prevent bat population declines due to human activities, while also mitigating risks of pathogen spillover from bats to humans.

## Figures and Tables

**Figure 1 microorganisms-08-00440-f001:**
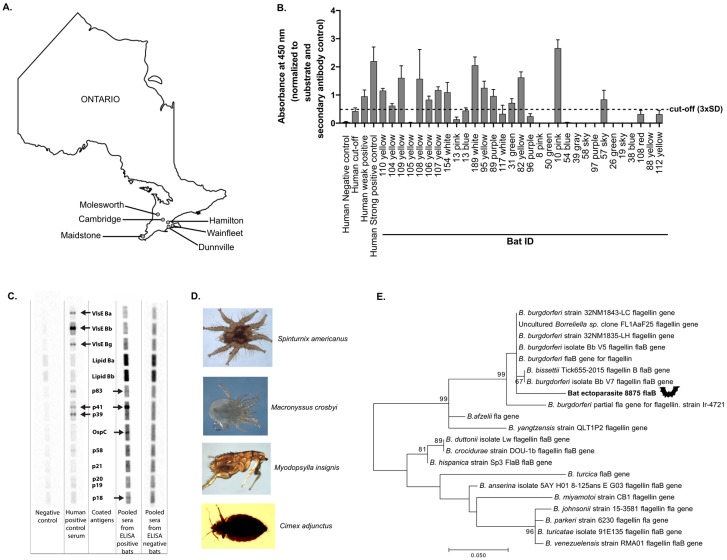
Detection of antibodies to *Borrelia* spp. in *Eptesicus fuscus* and the *flaB* gene segment in bat ectoparasites. We extracted sera from wild-caught and laboratory-bred *E. fuscus* to detect antibodies to *Borrelia*. **A**. Sites of bat collection. Bats were later kept in a captive research colony. **B**. Bar graph: antibody levels detected in *E. fuscus* serum samples using a commercial ELISA kit specific for anti-*Borrelia burgdorferi* antibodies. Human control sera were assayed in parallel. The cut-off for bat samples was set at 0.475 and is represented by the dotted line. **C**. Reactivity of pooled bat (*E. fuscus*) and control human sera with antigens from *B. burgdorferi, Borrelia afzelii*, and *Borrelia garinii*. See manufacturer’s manual for antigen details (EUROIMMUN, Germany). **D**. Ectoparasites collected from bats in Canada. **E**. Detection of *Borrelia burgdorferi flaB* gene fragment in a bat ectoparasite. The maximum likelihood tree (1000 bootstraps) showing the phylogenetic relationship of the *Borrelia burgdorferi flaB* gene segment detected in 1 of 142 bat ectoparasites that were collected between 2003 and 2019 is shown. The tree is drawn to scale, with branch lengths measured by the number of substitutions per site. The percentage of trees in which the associated taxa are clustered together is shown next to the branches. Values over 60 are shown. Evolutionary analysis was conducted in MEGA7. The following terms are used: SD (standard deviation); Ba (*Borrelia afzelii*); Bb (*Borrelia burgdorferi*); Bg (*Borrelia garinii*).

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
