# Peer review of "Seroprevalence in Bats and Detection of Borrelia burgdorferi in Bat Ectoparasites"

_microorganisms, 2020, doi:10.3390/microorganisms8030440_

Round 1

Reviewer 1 Report

Banerjee et al asked a very interesting question whether bats can be infected with Borrelia burgdorferi ss and whether they can be reservoir for this human pathogen.

Unfortunately, the study design had some issue. While the ELISA and the immunoblot provided promising data, the PCR and culture did not confirmed the results any of the bats studied. The only positive PCR data was found in one ectoparasites of over hundred parasites tested.

The PCR should have produced some confirmation but the author only tried one PCR target. There are well known for clinical and field samples that multiple PCR target needs to be included because they are a lot of different species for Borrelia spp and we do not know the exact species which can infect bats. The PCR /sequencing data from that ectoparasite does not mean that is the species which bats can harbor.

Therefore, to accept this manuscript the authors need to repeat the PCR with additional targets, such as 16S rDNA, pyrG etc….

One more question: while did not the human positive control sera procure a band for the OspC antigen?

Reviewer 2 Report

This is a well-written, interesting paper. The fact that Borrelia seroreactivity was detected in captive offspring (after the bats were treated to prevent the spread through ectoparasites) suggests vertical spread and/or possible contact transmission, and the presence of living infectious organisms. Various means of transmission to offspring could be more thoroughly discussed. Despite all the positive aspects of this manuscript, I do have some concerns that need to be addressed.

  1. Figure 1 is blurry and even when I zoom in I cannot see it in sufficient detail. It is horrible and needs to be in far higher resolution.

  1. The authors mention the role of bats in the enzootic cycle of Relapsing Fever (RF) in the first sentence of the manuscript, yet the testing for the detection of exposure to Borrelia spp appears to be targeted to Borrelia burgdorferi sensu stricto (Bbss) and B. burgdorferi sensu lato (Bbsl). The authors assume that their study also detects exposure to RF Borreliaspp (RFB). In actuality, serological cross-reactivity may be far more limited and serological detection of Bbsl reactivity does not necessarily mean that reactivity to RFB strains will be detected. [1-3] The presence of RFB may be under-appreciated in this study.

  1. ELISA. To detect Bbss, ELISA plates were coated with Bbss antigens. This may be an inadequate method to detect seroreactivity to Bbsl and RFB strains. Reactivity against sera reactive to RFB, and even perhaps to Treponema spp, would be useful. There are insufficient controls to show the specificity of this test to sera reactive to Bbsl and RFB. If these controls are not performed the fact that the tests may be limited in detection should be fully addressed. The ability to detect crossreactive antibodies must be more thoroughly explored and/or discussed.

  1. Immunoblots. To detect Bbsl, commercially available strips coated with antigens from Bbss, B. afzelii and B. garinii were used. Like the above, this may be an inadequate method to detect seroreactivity to RFB strains. Reactivity against sera reactive to RFB and even perhaps to Treponema spp would be useful. There are insufficient controls to show the specificity of this test to sera reactive to RFB. If these controls are not performed the fact that the tests may be limited in detection should be fully addressed. The ability to detect crossreactive antibodies must be more thoroughly explored and/or discussed.

  1. PCR was used to detect Borrelia DNA from a mite from one bat – from 1/142 ectoparasites collected. The DNA detection methodology was included under the immunoblot methodology. I would like to see a more detailed methodology on PCR, including what controls were used. The data in the figure is blurry, but the strain detected appears to most closely match DNA of B. burgdorferi. Was the primer used also capable of detecting RFB? There is no information concerning primer specificity. If the primer is specific for Bbss or even Bbsl it may be missing many Borrelia strains. Also, just because Borrelia DNA was detected from the mite, it does not mean that there was viable, transmittable infection present in the mite. This fact must be mentioned.

  1. While bats do carry many zoonotic diseases, including rabies, histoplasmosis, salmonellosis, and yersiniosis, bats play an important beneficial ecological role. They have a vital role in controlling insect populations, and therefore vector-transmitted diseases such as Zika virus and West Nile virus. In addition, bat populations are declining because of the presence of human activity, mostly by the spread of White Nose Syndrome. This paper underscores the need to better isolate bat colonies from human activity, both to protect humans from zoonotic diseases and to protect bats from human activities that cause population decline. The importance of protecting bat colonies should be included in the discussion.

References:

  1. Liu S, Cruz ID, Ramos CC, Taleon P, Ramasamy R, Shah J. Pilot Study of Immunoblots with Recombinant Borrelia burgdorferi Antigens for Laboratory Diagnosis of Lyme Disease. Healthcare (Basel). 2018 Aug 14;6(3). pii: E99. doi: 10.3390/healthcare6030099.

  1. Middelveen MJ, Shah JS, Fesler MC, Stricker RB. Relapsing fever Borrelia in California: a pilot serological study. Int J Gen Med. 2018 Sep 21;11:373-382. doi: 10.2147/IJGM.S176493. eCollection 2018.

  1. Shah JS, Liu S, Du Cruz I, Poruri A, et. al.Line Immunoblot Assay for Tick-Borne Relapsing Fever and Findings in Patient Sera from Australia, Ukraine and the USA.Healthcare (Basel). 2019 Oct 21;7(4). pii: E121. doi: 10.3390/healthcare7040121.

Round 2

Reviewer 1 Report

Thank you for the detailed feedback. It is very well known that PCR from clinical and field samples are very challenging that is why most studies are using multiple targets especially when it comes back negative from seropositive samples.  Acceptance of this manuscript requires positive PCR. Please run the samples with other published PCR protocols, I am sure it will work and make this study much stronger.

Author Response

We understand the reviewer’s concern but we respectfully disagree that using additional PCR approaches will yield a positive result simply because the organism is seropositive. We have tried two different targets on bat blood cells (16s rRNA and 5s-23s rRNA intergenic spacer) and all of our attempts to amplify these Borrelia targets using DNA extracted from bat blood cells were negative. We also tried to enrich for Borrelia by culturing bat blood cells in Borrelia-specific media, but we were unable to detect spirochetes by immunofluorescence microscopy and cpn60 total population sequencing. There are two scenarios that we have proposed in the discussion:

  1. Bat blood cells may not be a suitable sample to detect Borrelia. We have discussed the need for more invasive sampling.
  2. It is possible (and highly likely) that bats have cleared infection. The reviewer mentioned that multiple PCR approaches may yield a positive since the animals were seropositive. To our knowledge, seropositivity does not indicate an active presence of bacteria/infection. It is possible, and likely, that bats have already cleared infection. Thus, based on seropositivity, we cannot rationalize deploying an exhaustive PCR search to look for a bacteria that may not currently exist in the host. This would require sacrificing an animal without sufficient justification, which would not be approved by our animal care committee.

We believe that our manuscript reports a very intriguing phenomena that obviously requires further investigations, and extensive surveillance and mechanistic studies.

Reviewer 2 Report

The revised paper should be accepted and no further revisions are required. 

Author Response

Thank you.

Round 3

Reviewer 1 Report

Thank you for your feedback on the difficulties of the PCR. Please include this discussion in the manuscript so the readers can understand the different possibilities of failed PCR from bats' sera.

Author Response

We have included this information in the revised manuscript draft (lines 203-208).